# Amino Acid-Based Boron Carriers in Boron Neutron Capture Therapy (BNCT)

**DOI:** 10.3390/pharmaceutics15122663

**Published:** 2023-11-23

**Authors:** Juulia Järvinen, Herkko Pulkkinen, Jarkko Rautio, Juri M. Timonen

**Affiliations:** 1School of Pharmacy, Faculty of Health Sciences, University of Eastern Finland, 70211 Kuopio, Finland; 2Department of Technical Physics, Faculty of Science, Forestry and Technology, University of Eastern Finland, 70211 Kuopio, Finland; 3Drug Research Program, Division of Pharmaceutical Chemistry and Technology, Faculty of Pharmacy, University of Helsinki, Viikinkaari 5E, P.O. Box 56, 00014 Helsinki, Finland

**Keywords:** boron neutron capture therapy, BNCT, LAT1, amino acids, glioma, boron carriers, tumor targeting

## Abstract

Interest in the design of boronated amino acids has emerged, partly due to the utilization of boronophenylalanine (BPA), one of the two agents employed in clinical Boron Neutron Capture Therapy (BNCT). The boronated amino acids synthesized thus far for BNCT investigations can be classified into two categories based on the source of boron: boronic acids or carboranes. Amino acid-based boron carriers, employed in the context of BNCT treatment, demonstrate significant potential in the treatment of challenging tumors, such as those located in the brain. This review aims to shed light on the developmental journey and challenges encountered over the years in the field of amino acid-based boron delivery compound development. The primary focus centers on the utilization of the large amino acid transporter 1 (LAT1) as a target for boron carriers in BNCT. The development of efficient carriers remains a critical objective, addressing challenges related to tumor specificity, effective boron delivery, and rapid clearance from normal tissue and blood. LAT1 presents an intriguing and promising target for boron delivery, given its numerous characteristics that make it well suited for drug delivery into tumor tissues, particularly in the case of brain tumors.

## 1. Introduction

Cancer is a complex and deadly disease that continues to pose a significant global health challenge, causing millions of deaths worldwide each year [1,2]. Despite significant advances in treatment and research, it remains a challenging threat to public health. Although traditional therapies such as surgery, radiation, and chemotherapy continue to be prevalent in treating this devastating disease, new approaches are being explored to enhance treatment modalities. Among these promising treatments is Boron Neutron Capture Therapy (BNCT), a non-invasive technique that selectively destroys tumor cells using energetic alpha particles generated by neutron capture of boron. BNCT has shown potential for treating previously irradiated areas and tumors near critical organs like the brain stem or spinal cord [3]. However, despite the progress in cancer treatment modalities, high mortality rates for metastatic cancer still exist, highlighting the crucial need for ongoing research, especially in developing more targeted boron carriers for difficult-to-treat tumors such as gliomas, the most common primary brain tumors in adults.

BNCT offers a selective way to destroy malignant cells without harming normal cells [4,5]. By injecting a boron-containing compound into the body and exposing the targeted compound to a beam of low-energy thermal neutrons, a large dose gradient can be created between tumor cells and healthy cells, sparing normal cells [4,5,6]. BNCT has shown promise in treating a variety of difficult-to-treat tumors, including locally recurrent malignant gliomas [7], head and neck cancers [8,9], and recurrent lung cancer [10]. Currently, only two boron carriers, l-boronophenylalanine (l-BPA; l-**1** in Figure 1) and sodium borocaptate (BSH; **2** in Figure 1), are in clinical use, but they have limitations in terms of poor water solubility, lack of tumor selectivity, and undesired interactions with other biomolecules [11,12,13].

New boron carriers for BNCT have been developed since the 1950s, and a diverse set of approaches to carry boron into cells has been harnessed. Among the most recent are organic and inorganic nanocarrier-based solutions. Nanomedicine offers longevity and targeting possibilities, but toxicity and varying in vivo results on boron accumulation have challenged the development (Appendix A) [14]. Another strategy utilizing polyhedral borane structures, such as carboranes, may provide a high boron content relative to molecular weight and exhibit good stability [15]. However, challenges in the administration route to the tumor site, especially via biological barriers, still require resolving. Several biomolecules, like carbohydrates or nucleosides, conjugated with boron moieties like carborane or boronic acids may increase solubility and enable targeting to tumors. However, the transport mechanisms are not well understood yet. Currently, glucose conjugates have shown promise with tumors’ abundance of GLUT-transporters, but compounds may form undesirable micelles depending on the sugars utilized [16]. Similarly, nucleoside conjugation with boron agents has shown selectivity on tumor cells, but both accumulation and solubility issues have challenged the research [17].

Brain tumors present a significant challenge for effective treatment due to their location behind the blood–brain barrier (BBB) [18]. The BBB acts as a protective barrier, limiting the passage of most drugs into the brain and making it difficult to achieve adequate drug concentrations at the tumor site [19]. This challenge is further compounded by the aggressive and invasive nature of gliomas, which makes complete surgical removal difficult.

A few transporters with high enough transport capacity have been identified to overcome this challenge, offering promise for effective drug delivery to the brain. One such transporter is the large amino acid transporter 1 (LAT1; SLC7A5), which accepts various compounds as substrates, making it an attractive target for drug delivery [20]. LAT1 is highly expressed on the BBB, as well as the surface of cancer cells, including brain tumors, as they require amino acids for continuous growth and proliferation. Cancer cells are unable to synthesize essential amino acids adequately, and therefore, they must take up amino acids from their surroundings. LAT1 is a Na^+^-independent transporter protein that transports large neutral amino acids such as l-phenylalanine, l-leucine, and l-isoleucine from extracellular fluids into the cells [21,22]. While transient disruption in the supply of essential amino acids to the brain does not cause irreversible damage, the high expression of LAT1 on cancer cells offers the potential for targeted drug delivery to brain tumors [23].

Breakthroughs have occurred with the recent developments in hospital-based BNCT devices and new analytical methods for studying boron accumulation, distribution, and localization in cells in vitro and in vivo (for a recent review of in vivo models in BNCT, see references [24,25]). In addition, rapid progress in the research of LAT1 has revealed the vast potential of using amino acids in BNCT. Amino acid-based boron delivery agents may therefore represent the next breakthrough in treating brain tumors with BNCT [26]. This review will give an extensive historical perspective of the development of (amino acid-based) boron carriers. We also address the developmental challenges in utilizing amino acid-based boron carriers for BNCT as a treatment modality for different malignancies, including (brain) tumors located behind the uncompromised BBB. In particular, we focus on the promising potential of LAT as a target for boron carriers for BNCT. We will also consider some critical issues that must be addressed to increase the efficiency of this treatment modality, while there is an urgent need for improved boron carriers.

## 2. BNCT

BNCT was initially evaluated clinically in the 1950s, and since then, significant progress has been made via a wide range of clinical and non-clinical research efforts. Over 2000 cancer patients have been treated with BNCT globally so far, with a variety of diseases, including glioma [27,28], meningioma, head and neck cancer [9,29], lung cancer, breast cancer, hepatocellular carcinoma, sarcoma, melanoma, and a few other malignancies [30]. Approximately half of the treated patients had brain cancer, and one-third had head and neck cancer. Besides cancer therapy, research has been conducted to apply BNCT to counter infection caused by implanted devices non-invasively [31,32], and it was proposed to be utilized with boronated curcumin analogs to enhance the treatment of Alzheimer’s disease [33]. Despite the potential, BNCT has not become a standard method for radiotherapy yet. Until recently, the conduct of clinical BNCT required the use of a modified nuclear reactor. In the past, more than ten reactors were used for clinical BNCT worldwide, but they were inconvenient for clinical purposes [34]. For this reason, an emerging trend is to consider an accelerator-based neutron source for clinical BNCT [35,36]. The first clinical trials with accelerator-based BNCT have already been completed [37,38].

During a typical BNCT treatment (Figure 1), the patient receives an intravenous infusion of 250–500 mg/kg body weight of the ^10^[B]BPA-fructose complex (BPA-F) over 2–3 h [35,39,40]. Following this, the patient is exposed to thermal or epithermal neutrons for an hour, depending on the depth of the tumor. The absorption of neutrons by ^10^B activates a nuclear fission reaction where the formed labile ^11^B atom decomposes into a ^4^He particle and a lithium-ion (^7^Li^+^). These particles produce high-linear energy transfer (high-LET) within an extremely short distance (<10 μm, a diameter of a single cell) capable of damaging tumor cells containing an adequate amount of ^10^B, resulting in cell cycle arrest and apoptosis [41,42]. The intranuclear localization of ^10^B enhances the cell-killing effect because radiation is more likely to damage the Deoxyribonucleic acid (DNA) [43]. 

When using so-called thermal neutrons, which have an energy of <0.5 keV, the neutron capture cross-section of ^10^B is high (3838 barns, 1 barn = 10^−24^ cm^2^) compared to common elements presented in tissue (<33 barns) [44]. The penetration depth of the thermal neutrons is limited, and the maximum flux is approximately 2–3 cm from the skin surface [45]. Epithermal neutrons with higher energies (<10 keV) have been used to increase the neutron penetration depth to 6–8 cm from the skin surface. These neutrons release energy and become thermal as they scatter from hydrogen atoms along their path. This scattering sets the limit for the energy of the neutron beam used, and neutrons with over 10 keV energy increased the damage inflicted on normal tissue along the neutron beam path [46,47]. Therefore, the BNCT is suitable for shallow-seated tumors [35].

Despite the intensive research of recent decades, only two boron carriers (Figure 1), l-BPA and BSH, are currently in clinical use. BPA, a neutral amino acid derivative, is capable of selective accumulation in tumors via amino acid transporters that are highly expressed in tumors, such as LAT1, which is proposed as the primary transport route for BPA uptake [48]. Unlike BPA, BSH is an inorganic complex with charges, making it unable to pass through the intact BBB by passive diffusion. Therefore, ^10^B concentration in the normal brain is deficient after BSH administration. However, the BBB is disrupted in malignant brain tumors, allowing BSH to accumulate in tumors. Thus, BSH is mainly used to treat malignant gliomas where the BBB is impaired. Both BPA and BSH have limitations as boron carriers for BNCT. BPA has poor water solubility, only one boron atom/delivered molecule, and poor tumor selectivity over blood and normal tissue [11,12]. On the other hand, BSH lacks active targeting and uptake mechanisms and has an ionic nature, which may cause undesired interactions with other biomolecules in a biological context [13]. These challenges highlight the need to develop more efficient and specific boron carriers for BNCT.

## 3. Amino Acid-Based Boron Carriers for BNCT: A Journey through Development and Challenges

To improve the efficacy of BNCT, significant effort has been devoted to developing novel boron carriers with improved boron delivery into the cancer cells and favorable pharmacokinetic characteristics for clinical use. The novel boron carriers have to meet numerous requirements to be considered for clinical use. These include sufficient aqueous solubility, minimal systemic toxicity, a minimum cellular accumulation of 20−35 μg of ^10^B isotope/g of the tumor, high tumor/normal tissue (T/N) and tumor/blood (T/B) ratios, and a detailed understanding on the behavior of the delivery agents in a biological setting [49]. Also, the chemistry of boron-containing compounds presents a significant challenge to the synthesis of boron carriers. 

In general, three generations of boron carriers have been developed for BNCT, with occasional improvements to create an ideal boron carrier [50]. 

First generation: In the 1950s and early 1960s, inorganic boron compounds, e.g., boric acid, borax, and pentaborates, were synthesized and tested in clinical trials. However, poor tumor accumulation was observed, resulting in much lower concentrations of tumors compared to the brain [49].Second generation: The two most efficient and prominent boron compounds, BPA and BSH, emerged in the 1960s. These boron carriers had significantly less toxicity and persisted longer in animal tumors than other tested compounds. In addition, T/B and T/N ratios were shown to be higher than 1. Promising results in vitro and in vivo resulted in these boron compounds being transferred to clinical trials and eventually to clinical use [51].Third generation: Recently, boron clusters with high hydrolytic stability have emerged as tools for BNCT; however, their toxicity and lack of tumor selectivity limit their direct benefits in BNCT. There are a wide variety of intellectual developments in this regard, such as polyamines, unnatural amino acids, peptides, proteins, nucleosides, sugars, porphyrins, antibodies, liposomes, and nanoparticles conjugated with either BPA, BSH, or other boronated compounds to develop a better delivery system [52].

### 3.1. Prehistory—Treatment of Melanoma with BPA and Its Isomers

The first boronated amino acid, racemic 4-boronophenylalanine or *p*-boronophenylalanine (BPA, **1**), was published in 1958 by Snyder et al. [53]. In the early years, boronated compounds’ accumulation mechanism was not understood. In 1960, it was observed that the penetration of brain and brain tumors with a series of aromatic boronic acids, including BPA, is not straightforward and could not be explained solely by lipophilicity [54]. In vivo studies in 1961 with several boronated compounds showed other compounds besides BPA to be the most promising [55]. In the 1980s, the role of phenylalanine and tyrosine in melanin synthesis in melanoma was proposed to improve the uptake of BPA, and the study of BNCT focused mainly on melanoma. Several researchers studied the uptake of BPA in vitro and in vivo and the efficiency of BPA for treating melanoma in vivo (See Appendix A) [46,47]. Despite intensive research, it was not initially clear how BPA accumulated in cellular compartments in vivo or in vitro nor its exact role in melanin biosynthesis, e.g., in vitro using human uveal melanoma cell line MK-T (Appendix A, entry 6) and in vivo xenografts (Appendix A, entry 1; Appendix A, entry 1) [56,57]. 

However, in 1998, enhanced melanin biosynthesis was shown to significantly increase boron accumulation in an amelanotic mouse melanoma cell line A1059 without increasing the metabolic rate [58]. In the study, human tyrosinase-cDNA was transfected into amelanotic melanoma cell line A1059, leading to a novel melanotic cell line, TA1059. This new melanin-producing cell line with A1059 and mouse melanoma cell line B16-F1 were studied for their boron accumulation in vitro (Appendix A, entries 4, 7, and 8). Also, it was found to respond better to BNCT in vivo than the primary A1059 cells (Appendix A, entry 2). Two other isomers of BPA, 3-boronophenylalanine (*o-*BPA, **3** in Figure 2) and 2-boronophenylalanine (*m-*BPA, **4**), have been studied using an in vivo melanoma model (Appendix A, entry 2). Disappointingly, both isomers showed lower boron accumulation in the tumor compared to BPA [59].

After the role of melanin synthesis in BNCT of melanoma was established, BPA emerged as the leading boron carrier. After encouraging results with melanoma, interest in also curing brain tumors, especially gliomas, using BNCT, emerged in the 1990s. Studies showed BPA efficiency to accumulate in brain tumors in vivo and in vitro (Appendix A). Also, subcellular localization in vitro and effect of BBB disruption into BPA accumulation in vivo were studied using T98G human glioblastoma and F98 rat glioma models (Appendix A, entry 11; Appendix A, entry 1; Appendix A, entry 3) [60,61].

Recently, studies of different melanoma cell lines have highlighted their varying sensitivity to BNCT. In 2015, Rossini et al. showed that the survival of M8 cells after treatment with γ-rays, neutron beam contaminated with γ-rays, and neutron beam after [^10^B]BPA treatment was significantly lower than the survival of Mel-J cells (Appendix A, entries 1 and 2) [62].

### 3.2. The Challenge of Non-Invasive Detection—Positron Emission Tomography (PET) and Magnetic Resonance Imaging (MRI)

The lack of real-time, non-invasive monitoring of boron concentration in tumors is one of the major challenges in clinical BNCT. However, the development of positron emission tomography (PET) and magnetic resonance imaging (MRI) has opened new prospects for monitoring the accumulation of targeted compounds in vivo. Several fluorinated BPA derivatives have been developed for combined PET and BNCT [63]. As the half-life of PET-active radioisotope of fluorine (^18^F) is relatively short (about 20 min), the syntheses and initiative biological studies were carried out using non-radioactive fluorine. In 2006, Hattori et al. [64] studied in vitro the boron accumulation using [^10^B]-2,6-difluoro-4-boronophenylalanine (**5** in Figure 3) in a rat glioma cell line C6, a human squamous carcinoma cell line KB, and a human epithelioma cell line HeLa (Appendix A, entry 1; Appendix A, entries 1 and 4, respectively). The C6 cells showed twice as high boron accumulation for [^10^B]BPA than that after dosing **5** [65]. In contrast, the KB cells showed similar boron accumulation for **5** and [^10^B]BPA. 

To evaluate boron-containing PET imaging agents, radiolabeled [^18^F]-2-fluoro-4-boronophenylalanine (**9**) was synthesized and studied for biodistribution in healthy mice (Appendix A, entry 1) by Ishiwata et al. [66]. Also, the boron accumulation was shown to correlate with both DNA synthesis and melanin production in three human melanoma models (Appendix A, entry 3) [67]. In a recent study, the correlation between the accumulation of BPA and **9** was shown in several cell lines, including the U-87 MG, LN-229, LN-18, U-118 MG, and U-251 MG human glioblastoma cell lines, FaDu squamous cell carcinoma cell line, and A-253 epidermoid carcinoma cell line (Appendix A, entries 4, 7, 11–13; Appendix A, entries 2 and 5). The biodistribution results for U-87 MG, LN-229, U-118 MG, U-251 MG, FaDu, and A-253 were also confirmed in an in vivo xenograft model using [^18^F]**9** and [^14^C]BPA (Appendix A, entries 2–5; Appendix A, entries 2–4, respectively) [68]. For a recent review on radiofluorination, see the excellent review article by Deng and Yu [69].

Fluorine-NMR (F-NMR) has also been shown to detect the uptake of fluorine-containing boronated amino acids **5** and **6** from a human melanoma cell line, Ihara (Appendix A, entry 9) [64]. The results of **5** and **6,** and their corresponding alcohols, **7** and **8**, were of similar magnitude as the acid derivatives. Furthermore, the boron accumulation increased when preloading cells with l-BPA, l-tyrosine, or l-DOPA before incubation with **9**, as demonstrated in three different cell lines (human hepatocellular carcinoma cell line HuH-7, human colorectal cell line CaCo-2, and mouse melanoma cell line B16-F1) (Appendix A, entries 6 and 7; Appendix A, entry 6, respectively) [70,71].

In a recent study, a negatively charged trifluoroborate group was introduced to mimic the carboxyl group in natural tyrosine, resulting in the creation of fluoroboronotyrosine (FBY, **10**). The compound showed remarkably higher metabolic stability toward H_2_O_2_ and it was transported into melanoma cells dose-dependently (Appendix A, entry 6). In vivo xenograft distribution and neutron capture studies (Appendix A, entry 4; Appendix A, entry 4) showed higher accumulation into tumors and higher survival rates for FBY than for BPA [72].

### 3.3. The Challenge of Boron Accumulation—Polyhedral Boron Cages for Higher Load

Because the efficiency of BNCT is primarily dominated by ^10^B accumulation inside malignant cells, boron carriers with more than one boron atom have drawn attention since the early BNCT research. Boron cage-containing compounds have also received considerable attention due to their higher catabolic stability and lower toxicity, leading to the development of several applications for BNCT. One example of these high boron load compounds is BSH (**2**). The first synthesized amino acid derivative with a polyhedral borane was (*o*-carboranyl)-alanine (*o*-CAR or *o*-CBA, **11**), synthesized in the 1970s. It was designed to mimic phenylalanine as the carborane cage was thought to occupy the same space as a rotating phenyl ring of phenylalanine [73,74]. In the initial development of carborane-containing amino acids, dicarba-closo-dodecaboranes were utilized. These compounds possess an icosahedral closed polyhedron structure with ten boron atoms and two carbons positioned either at the ortho (**I** in Figure 4) or meta (**II** in Figure 4) positions. Other boron-rich polyhedral structures utilized in boron carriers are closed polyhedrons with more than ten boron atoms and seven boron-containing structures lacking one vertex (*nido*-form, **III** in Figure 4). Only scarce examples of *nido*-structures were prepared (e.g., **12**), and none were biologically evaluated. Instead, successful clinical use of BSH has inspired studies of twelve boron-containing BSH-like *closo-*dodecaborate-ion (**IV** in Figure 4) containing structures, e.g., **13**–**15**. 

Boron accumulation using *o*-CBA (**11**) was studied in vitro and in vivo to treat melanoma and glioma (entries 11, 5, and 6 in Appendix A, respectively). Disappointingly, the boron accumulation in vitro was less than half of that observed with l-BPA. Also, in vitro results showed fast boron removal after transferring cells into a boron-free medium: no boron was detected from cells after 60 min at a boron-free medium. However, in vivo results using an implanted melanoma model demonstrated that **10** exhibited a higher T/B ratio compared to l-BPA. However, accumulation of **10** into the tumor was not detected with F98 glioma implanted rats [75]. Other derivatives containing boron clusters included, for example, different spacers between amino acid moiety and the carboranyl, e.g., alkyl (**16**) or aryl (CBT, **17** and CBTA, **18**), but their in vitro or in vivo fate was not reported [76,77]. In 1998, a promising water-soluble amino acid derivative, DAAC-1 (diol-amino acid-carborane, **19**), was prepared and studied for its cell uptake, toxicity, and radiation effects on a human glioma cell line, U343 (Appendix A, entry 14) [78]. Studies showed that **19** was nontoxic (Appendix A, entry 9) at the used concentrations, and the boron accumulation was time- and dose-dependent. Also, in neutron capture studies with cell suspension, the increase in survival after **19** incubations was statistically significant. However, due to low boron accumulation in the cell nucleus and overall fast removal from the cells, **19** was not considered a highly potential boron carrier.

In the 2010s, BSH-derived amino acids **13**–**15** were synthetized and evaluated for their efficiency in BNCT. From the compounds, in vitro studies showed the highest boron accumulation in C6 brain tumor cells for **13** and **15** (Appendix A, entry 2). While in melanoma B16 cells (Appendix A, entry 1), all three derivatives showed better boron accumulation than l-BPA. In vitro BNCT survival studies (Appendix A, entries 3–5) showed that **14** and **15** had the lowest survival rate in C6 and B16 cells, while **13** and **14** showed the highest accumulation in tongue squamous cell carcinoma (SAS) cells (Appendix A, entry 3). The best response in survival studies was found in C6 cells, with a survival rate almost ten-fold smaller than in other cell lines [79]. In vitro localization studies with C6-cells showed **13** accumulating into the cell fringe. The compound **13** also showed increased cell viability after 24 and even more after 48 h. Additionally, the compound significantly affected cell gene expression, particularly genes associated with the cell cycle [80]. Recently, studies of **20** showed that it is rapidly taken up by glioma U87 cells and is retained in cells for over 48 h. In neutron capture studies, **20** showed higher time- and dose-dependency than l-BPA (Appendix A, entry 6). Furthermore, polyols have been attached to unnatural carboranyl-bearing amino acids, e.g., **21** and **12** [81]. To the best of our knowledge, detailed biological data for these two interesting compounds have not been published.

### 3.4. The Challenge of Water Solubility—Complexes with Saccharides and Incorporation of Hydroxyl Groups

The low solubility of amino acid-based boron carriers under physiological pH conditions has also hindered the advancements of BNCT (Table 1). The pKa values for BPA were determined to be 2.46, 8.46, and 9.76. This indicates that at physiological pH (around 7.4), the most abundant species is the most insoluble zwitterionic form of BPA [82]. At first, the solubility of BPA was enhanced by using hydrochloric salt (BPA•HCl). However, the pH of a 0.1 M solution of BPA•HCl is highly acidic (about 1.5), making the injection painful for the patient. Monosaccharides are known to form complexes with boronic acids, and the complex formation of *p*-BPA has been studied with different monosaccharides, e.g., galactose, glucose, mannitol, and fructose. All of the complexes are highly water soluble. The BPA complex with fructose has been shown to be the strongest, and its properties and stability have been studied using various methods, e.g., ^11^B-NMR, ^1^H-NMR, and ^13^C-NMR [83,84]. As a result, fructose has become a commonly used component in other boric acid-based boron carriers, distinct from BPA.

Another studied strategy to improve the water solubility of boron carriers is to alter the structure of amino acids by incorporating hydroxyl groups. This increases the hydrophilicity of the molecule and consequently increases its water solubility. One example is boronophenylserine (**22** in Figure 5), in which a hydrogen of the alkyl chain of BPA has been substituted with a hydroxyl group [85]. An additional hydroxyl group increased the solubility of the molecule to almost five-fold (Table 1). Additionally, polyols containing multiple hydroxyl groups, such as those found in structures (**23**–**25**), have also been investigated for their potential to improve the water solubility of boron carriers. As hypothesized, the water solubility increased as the number of hydroxyl groups increased (Table 1), but the results of in vitro studies using human melanoma cell line B-16 and a non-cancerous cell line TIG were disappointing (Appendix A, entry 2; and Appendix A, entry 8). The boron accumulation after incubation with **23** was lower than that of BPA and decreased as the number of hydroxyl groups increased [12]. Another study concentrated on attaching a polyol moiety to 1,3-decaborane of compound **26** [86]. Unfortunately, biological studies for this compound are lacking. 

### 3.5. The Challenge of Targeting—Pure Enantiomers and Novel Amino Acid Scaffolds

Targeting cancer cells is a challenging endeavor because cancerous cells are derived from the same origin as the healthy cells in the body. Various strategies have been attempted to address this challenge in developing amino acid-based boron carriers. One such strategy is based on the asymmetric nature of biological systems, and it utilizes asymmetric amino acids and their nutritional pathways. The earliest indication of the higher in vivo uptake of l-BPA compared to D-BPA was already observed in 1987 (Appendix A, entry 6) [46]. Development in asymmetric synthesis during the 1990s resulted in several enantiopure carboranyl-containing amino acids, e.g., l-**27**, l-**11,** and l-**16** (Figure 6) [87,88,89,90]. The progress of asymmetric synthesis of l-BPA was reviewed extensively in 2008 by Sivaev and Bregadze [91].

In 2004, an α-methyl-BPA (**28**) was synthesized after it was observed that α-methyl amino acids accumulate more selectively in cancer cells compared to their respective non-α-alkylated derivatives [92]. Recently, catalytic reactions have been used to prepare l-α-methyl-BPA and l-**28** [93]. Unfortunately, their biological evaluation has not been published to our best knowledge.

The development of amino acid-based BNCT took a remarkable leap after the discovery that cyclic amino acids accumulate in tumors more selectively than BPA [94]. In the early 2000s, several studies focused on using cyclic unnatural amino acid-based boron carriers (Appendix A, entry 7) [95,96,97,98]. The key invention leading to these innovations was the use of a particular synthetic method to prepare cyclic 1-amino carboxylic acids. One of the most promising compounds was *cis*-1-amino-3-boronocyclopentane-carboxylic acid (*cis*-ABCPC, *cis*-**29**) studied both in melanomas and gliomas (Appendix A, entry 3; Appendix A, entry 15; Appendix A, entries 8 and 9; Appendix A, entries 7 and 8) [99,100,101]. Also, several other compounds were synthesized and evaluated (see Figure 7 below) with different ring sizes, such as four, six, and seven (**30**–**33**) [96,100]. Additionally, different alkyl chains were employed as linkers between the cyclic backbone and the boronic acid moiety, including propyl-ether or ethyl (**34** and **35**) [102], as well as aromatic linkers as seen in **36**—**39** [103]. Unfortunately, most of these compounds failed in early in vitro or in vivo studies.

The combination of ACBC scaffold with BSH resulted in the creation of compounds *cis*-**40** and *trans*-**40**. In vitro studies showed higher boron accumulation in C6, A172, and U81 brain tumor-derived cells expressing LAT1 compared to l-BPA. However, this enhanced accumulation was not observed in U251. Additionally, these studies revealed higher boron accumulation in F98 cells than that observed for BSH but lower than for BPA (Appendix A, entries 3, 9, 16–18; Appendix A, entry 5) [104,105]. The in vivo study (Appendix A, entry 9) showed that the racemic compound was not able to cross the BBB. However, employing the CED (convection-enhanced delivery) technique, higher boron concentration in the tumor was detected using **40** compared to BPA or BSH. In vivo BNCT treatment studies were also carried out, and the results correlated with in vitro studies.

## 4. Discussion

BNCT has faced several obstacles that have hindered its widespread adoption. These challenges include limited experience with drug usage, difficulties in patient recruitment, a shortage of medical neutron sources, and other contributing factors. Consequently, both the development and clinical application of BNCT have progressed relatively slowly [106].

During the past decades, a wide variety of amino acid-based boron carriers have been synthesized, and many of them have also been studied in vivo and/or in vitro. Several of the synthesized structures are derived from natural amino acids, e.g., BPA and its derivatives. Also, structures mimicking natural amino acids and bioisosteres have been studied (e.g., *o*-(carboranyl)-alanine mimicking phenylalanine). However, the development of synthetic methods has led to the synthesis and biological testing of novel unnatural amino acid structures discussed above. In general, the challenge of chemical synthesis rises when the number of stereocenters increases, which might hinder the development of unnatural cyclic structures. Nevertheless, promising results from compounds like **40** could lead to the further development of entirely novel scaffolds for more selective boron carriers. 

As mentioned earlier, for a compound to succeed in BNCT, it must meet several requirements. A critical consideration, even in the early stages of development, pertains to achieving sufficient aqueous solubility [51]. It is worth noting that poor solubility has been identified as a potential limitation to the clinical success of certain boron carriers. Like BPA, which suffers from restricted solubility, BPA-F has made significant steps in enhancing its solubility and biocompatibility. However, the journey toward finding an optimal compound for BNCT remains challenging. 

A major reason for the poor solubility of all amino acid-based compounds is their ability to form a zwitterion, where the amino group is protonated, and the acid group is deprotonated while the total charge of the molecule is zero. One studied approach to enhance the water solubility of amino acid-based boron carriers is the modification of structures by turning the carboxylic acid into amide and incorporating hydroxyl groups, as exemplified by compounds **23**–**25**. These alterations increase the molecules’ hydrophilicity, subsequently improving their water solubility. However, it is important to note that although the solubility is increased, this approach has faced limitations during in vitro testing. The in vitro boron accumulation has been shown to diminish as the size of the amide group increases [12]. Reasons for the decrease might be related to the loss of active or passive transport.

Comparison of the research findings can be somewhat complex due to the utilization of various cell lines, each with its distinct characteristics. This substantial heterogeneity within the different cell lines, including those representing distinct cancer subtypes like glioblastoma, increases the complexity when attempting to draw comparisons between results. This diversity in cell lines is also evident in this review, as studies have extensively utilized numerous cell lines. The main reason for the diverse behavior of cell lines lies in the differential expression levels of genes, which in turn result in varying levels of proteins, such as amino acid transporters like LAT1.

The diversity in the presentation of results across studies poses a challenge for meaningful comparisons. Adopting a standardized method for measuring boron accumulation in cells and presenting results would significantly enhance the ability of researchers to compare findings systematically. Consistency in reporting not only facilitates clearer comprehension but also simplifies the comparative analysis of results across different studies.

### 4.1. The Role of LAT1 in Cancer Metabolism and Its Potential as a Therapeutic Target

To fully unlock the potential of BNCT as a targeted cancer treatment, the development of more specifically targeted boron carriers is urgently needed. One promising transporter of interest in this regard is LAT1, which has the capability to transport naturally occurring substrates and substrate-related compounds [22]. LAT1 plays a critical role in cell proliferation and is frequently overexpressed in various cancer cells, making it an attractive candidate for targeted delivery of amino acid-derived boron carriers as well as other anticancer drugs, prodrugs, and PET probes for cancer diagnostics [107,108,109]. Significantly, LAT1 is highly upregulated in human tumors, particularly in gliomas, where its expression can be 40- to 400-fold higher than in normal brain tissue [107]. This increased expression of LAT1 in human tumors, along with its prominent presence in the BBB and distribution on both the luminal and abluminal sides of brain capillary endothelial and parenchymal cells [110], establishes LAT1 as a compelling therapeutic target for brain tumor treatment. 

The BBB, acting as a diffusion barrier, presents a significant challenge in delivering drugs to brain tumors. The BBB consists mainly of brain endothelial cells, a specialized basal membrane rich in pericytes and astrocytes, which tightly regulate the movement of ions, molecules, and cells between the blood and the brain [111,112]. This highly regulated barrier serves as a natural defense mechanism against xenobiotics and pathogens, making drug delivery to brain tumors problematic [107].

However, LAT1 possesses several properties that make it well-suited for potential drug delivery across the BBB and into the brain [23]. It exhibits a large maximal transport capacity and moderate binding affinity, enabling rapid exchange across the BBB and short half-times for high-affinity substrates [113]. LAT1 accepts a range of amino acid-related compounds with higher affinities than endogenous substrates, including drugs like l-dopa, melphalan, triiodothyronine, thyroxine, gabapentin, and baclofen [22,114,115]. Importantly, the disruption of the brain’s amino acid supply resulting from LAT1 inhibition is considered safe and does not cause irreversible brain damage.

Extensive studies within the scope of BNCT have focused on malignant brain tumors, including both newly diagnosed and recurrent cases [30,116,117,118]. These tumors possess the ability to infiltrate the surrounding normal brain tissue, emphasizing the importance of achieving selective accumulation of boron atoms in tumor cells while sparing normal cells. In this context, the expression of LAT1 in gliomas has garnered attention, as it has been found to be higher in infiltrating glioma cells compared to cells located in the central region of the tumor [119]. Moreover, LAT1 expression levels in gliomas have been found to increase with tumor stage and grade, both within the glioma cells and at the BBB [120]. These observations suggest that targeting LAT1 with amino acid-based boron carriers could enable selective accumulation in tumor cells.

Considering substrate specificity and targeting, LAT1 predominantly transports l-amino acids with bulky or branched side chains [121]. Computational modeling indicates that an aromatic amino acid with a free carboxyl and an amino group is well-suited as a LAT1 substrate [22]. Hydrophobicity also plays a crucial role in the binding of amino acid side chains to the substrate binding site of LAT1, contributing to the stability of substrate binding [122,123]. However, there is relatively limited knowledge regarding the role of incorporated boron in LAT1 substrates that are typically devoid of boron. Computational docking simulations suggested that **10**, tyrosine, and BPA might exhibit similar interactions with LAT1 (Figure 2) [72].

The functional LAT1 is a heterodimeric complex consisting of a light chain (LAT1 and SLC7A5) and a heavy chain (4F2hc and SLC3A2) that are covalently linked to each other [121]. The heavy chain stabilizes the dimer and facilitates its translocation to the plasma membrane for effective transport. Studies have demonstrated that the inhibition of LAT1, but not 4F2hc, leads to significant tumor growth arrest, highlighting the role of LAT1 in cancer cell growth and its potential as a molecular target for anticancer therapeutics [124]. However, LAT1 exhibits overlapping substrate specificity with other amino acid transporters, such as LAT2 [125], which presents challenges in developing LAT1-selective compounds for targeted delivery.

In the context of BNCT, boronated amino acids like BPA have been investigated for their retention mechanisms and uptake pathways [47,126]. LAT1 has been identified as one of the primary pathways for the cellular uptake of BPA, although other transporters, such as ATB^0,+^, and LAT2, have been found to transport BPA with affinities comparable to their endogenous substrates [4,48]. The contribution of LAT1 to BPA uptake is particularly significant at lower concentrations, while the role of ATB^0,+^ becomes more prominent at higher concentrations. Understanding the transport mechanisms of boron carriers is crucial for optimizing their accumulation in tumor cells. Furthermore, the efficacy of BNCT can be affected by local hypoxia, which reduces the uptake of BPA in cancer cells and subsequently decreases the cytotoxic effect following neutron irradiation [127]. Local hypoxia often arises from the rapid growth of tumors and inadequate vascularization, posing a significant challenge to the therapeutic potential of BNCT [128]. Nonetheless, the distribution and levels of LAT1 may vary according to the specific tissue in the human body. This variability may, in part, contribute to the differences in boron uptake observed among individual patients and various tumor types. In addition to variations in the quantity and distribution of LAT1 expressions, the complex genomic and histologic diversity within tumors can also contribute to the variability in substrate uptake by tumor cells.

In summary, the development of specifically targeted boron carriers, harnessing the transport capabilities of LAT1, holds tremendous promise for enhancing the selectivity and efficacy of BNCT as a cancer treatment. Targeting LAT1 enables the selective accumulation of boron carriers in tumor cells, particularly in the context of malignant brain tumors. Overcoming the challenges posed by the BBB, optimizing substrate specificity, and addressing issues such as local hypoxia are crucial considerations in the development of effective LAT1-targeted therapies for BNCT.

### 4.2. Current State of Amino Acid-Based Boron Carrier Development

In recent years, only a few novel amino-acid-based boron carriers have been developed for BNCT. Until now, most of the novel compounds have failed either in vitro or in vivo studies due to poor selectivity, fast elimination, or low water solubility. Recently, novel approaches to incorporating boron into a natural amino acid, tryptophan, have been developed [129]. Most recently, a series of protected tryptophan derivatives **41**–**46** (Figure 8) were synthesized and tested for their capability to accumulate boron into tumor cells and effects on LAT1 expression (Appendix A, entries 5, 8 and Appendix A, entry 9) [130]. 

A new strategy to incorporate boron into the amino acid scaffold has also been developed. This method is based on complex formation with amino acid and 9-borabicyclo [3.3.1]nonane (BBN), giving the compounds **47**–**50**. The compounds were proposed to be incorporated into melanin based on the melanin biosynthesis mechanism. Although increased tyrosinase activity-related oxygen consumption was observed, none of the complexes **47**–**50** showed remarkably higher boron accumulation than l-BPA-F in SK-23-Mel melanoma cells or MCF-7 breast cancer cells in vitro (Appendix A, entry 12 and Appendix A, entry 10, respectively). The boron accumulation of **48** was significantly higher in melanoma cells than in MCF-7 cells [131]. The background boron levels were exceptionally high, ranging between 6 and 12 µg/L, which raises a question regarding the potential contribution of boron from alternative sources. One plausible example is borosilicate glassware, frequently employed in laboratory settings.

The third remarkable field of a recent study of novel compounds is the re-evoked interest in 3-boronophenylalaninen (*m*-BPA **3**) [132]. In the re-evaluation of **3**, its water solubility was shown to be remarkably higher than BPA (125 g/L vs. 0.72 g/L), but in vitro and in vivo studies did not show remarkable differences in boron accumulation in four different tumor cell lines (B16F10 mouse melanoma, T3M-4 human pancreatic adenocarcinoma, A549 human lung carcinoma and U-87 MG human glioblastoma (Appendix A, entry 5; Appendix A, entry 11 and 12; Appendix A, entry 6, respectively)) or statistically significant difference in biodistribution in vivo studies (Appendix A, entry 10) except higher accumulation of boron into kidneys when using **3** or its fructose complex. In a very recent research paper accepted during the writing process of this review, 3-borono tyrosine (**51**) inspired by the above-mentioned research has been synthesized and evaluated in vivo and in vitro [133]. The compound was also studied for its solubility by dissolving its hydrochloric salt into water and neutralizing this solution with a 10 M NaOH solution. Via this method, the compound showed solubility up to 250 mg/mL. However, boronic acids are Lewis’ acids and are thus capable of binding a hydroxyl ion from aqueous solutions in an equilibrium reaction that is enhanced in basic solutions. The scavenge of a hydroxyl ion leads to a tetrahedral borate ion whose dissociation might be hindered by structural stabilization. The structure of **51** was not determined after treatment, and it is not clear if there has been incorporation of hydroxyl ion into the boron. The biological evaluation of **51** in vitro with a wide variety of tumor cell lines showed higher boron accumulation, and diminished boron accumulation after the selective inhibition of LAT1-proposed LAT1-mediated transport. Despite higher levels of accumulated boron, higher V_max_, and longer retention of boron in cells in in vitro studies, the in vivo studies did not show a remarkable difference in biodistribution of **51** and l-BPA when using similar concentrations. However, the higher solubility of the compound allowed the use of doses up to 800 mg/kg body weight. With the highest doses, the boron accumulation into the tumor was improved, but also the accumulation of boron into the pancreas and kidney was remarkably higher compared to l-BPA. 

While only a limited number of novel molecules have been invented in recent years, several studies have focused on improving the use of existing amino acid-based boron carriers, e.g., l-BPA. One studied approach is to avoid the challenge of BBB by administrating the boron carrier directly into the cerebrospinal fluid. Studies with a rat brain tumor model showed that boron accumulation from cerebrospinal fluid peaked faster compared to i.v. administration. Also, the accumulation of boron and T/N ratios were similar to i.v. administration, even the dose of BPA was around 1/10 of that used in i.v. [134,135].

New methods to detect boron have also emerged recently. Among fluorescence sensors, novel compounds (**52**–**54**, Figure 9) have been developed based on a scaffold developed by Hattori et al. [136]. While **52** exhibits only a faint fluorescence, compound **54** achieved faster fluorescence after complexation with boronic acid. Also, **53** showed a considerably higher detection limit (0.82 µM) compared to **52** (65.3 µM) and **54** (6.01 µM). Fluorescence is easy to measure, and these new sensors could also be used inside the living cells, allowing for the real-time monitoring of boronic acid accumulation [137,138].

In addition to new measuring methods, more realistic in vitro models have emerged, with 3D cell models such as spheroids taking center stage [139,140]. These sophisticated models offer a valuable platform for evaluating boron accumulation within complex systems, moving beyond simple cell monolayers. Spheroids, in particular, have showcased their immense potential in probing the efficacy of BNCT across diverse microenvironments. Their application not only addresses the challenges associated with measuring intracellular boron accumulation in 3D models but also promises significant enhancements in our fundamental comprehension of BNCT. Furthermore, these innovative methods hold the promise of advancing the assessment of radiotherapy treatment efficacy and expanding BNCT’s clinical applications. Most notably, from an ethical standpoint, they offer the potential to reduce the reliance on animal experimentation in research projects.

## 5. Conclusions

The development of amino acid-based BNCT has been closely tied to the evolution of boron carriers, techniques, and methods over time. In the past, the lack of readily available neutron sources posed a significant obstacle to the widespread adoption of BNCT as a cancer treatment. However, with the recent installation of accelerator-based facilities for BNCT, progress is being made toward establishing its clinical use. Despite decades of research, over one hundred academic papers, and tens of failed boron carriers, a superior amino acid-based boron carrier for BNCT has not yet been found. Although boron carriers such as BPA and BSH have been used in clinical trials for over 60 years, they do not offer optimal therapy due to limited tumor specificity and insufficient delivery of ^10^B into tumors. Since the 1990s, BNCT research has predominantly concentrated on brain tumors, particularly high-grade gliomas, with fewer investigations conducted on melanomas and recurrent head and neck tumors. It is worth noting that other challenging cancers could potentially benefit from BNCT, either due to current treatment modalities being anatomically unsuitable or the absence of efficient treatment alternatives. Improving targeting and boron accumulation is essential for BNCT to reach its full potential.

Effective boron carriers for BNCT should have both tumor specificity and high ^10^B delivery to tumor cells, exceeding the amount required for tumor detection. The agent should deliver around 20–35 μg of ^10^B per gram of tumor, with high T/N and T/B ratios and be rapidly cleared from normal tissue and blood. However, developing amino acid-based boron carriers has been challenging due to hurdles in non-invasive detection, effective and selective accumulation in tumor cells, and poor water solubility. 

Despite the challenges in developing novel boron carriers, the research on BNCT has provided valuable insights into cancer biology, cellular internalization of amino acids, and the potential of BNCT as a cancer treatment modality. In particular, BNCT shows promise as a treatment for gliomas resistant to traditional X-photon irradiation. To achieve therapeutic effects, it is essential to accumulate sufficient ^10^B atoms in the tumor, which can capture epithermal neutrons to trigger the nuclear fusion reaction and destroy tumor cells, for clinical BNCT aimed at gliomas or other tumors shielded by the BBB, l-BPA and BSH have already secured their position. However, it is hoped that in the near future, more efficient amino acid-based boron carriers will become available in clinical settings to overcome the existing limitations. 

## 6. Future Directions

Indeed, the design of appropriate drugs for BNCT targeting LAT1 requires careful consideration and a deep understanding of the specific requirements for effective treatment. This includes considerations such as the expression levels of LAT1 in different types of cancer cells, the affinity of boron carriers for LAT1, the stability of the boron carriers in vivo, and the potential effects of local hypoxia on boron uptake. Further research and development are necessary to understand these requirements fully and to design boron carriers that can effectively utilize LAT1 for targeted delivery to cancer cells. Ultimately, a thorough understanding of the underlying mechanisms and factors influencing BNCT efficacy will be critical in advancing the development of effective and targeted cancer treatments.

To use LAT1 for effective drug delivery, it is essential to design and develop compounds that can effectively compete with the high concentrations of amino acids in the blood circulation [110]. One strategy for addressing this challenge is to create compounds with a higher affinity (K_m_) for LAT1 than the naturally occurring amino acid substrates. To achieve this, it is crucial to understand better the substrate specificity and the interactions between ligands and the transporter. By gathering this knowledge, researchers can develop compounds that can be efficiently transported by LAT1, potentially leading to improved drug delivery and therapeutic outcomes.

PET stands out as the most promising among the available methods for boron monitoring. This method holds immense potential as it not only enables imaging of the spatial distribution of ^10^B before treatment but also allows for real-time monitoring, thereby enhancing the effectiveness of the BNCT treatment [141]. Amino acid PET has several advantages and has been evaluated as a modality to complement anatomic MRI in neuro-oncology [142]. The PET task force of the Response Assessment in Neuro-Oncology Working Group highlights the outstanding clinical value of amino acid PET for glioma patients, justifying its widespread use at all disease stages. Similarly, the PET/Response Assessment in Neuro-Oncology Working Group has published recommendations for using amino acid PET in patients with brain metastases.

Amino acid PET tracers generally offer good T/B contrast, making them easily interpretable [143]. Standardization of image acquisition and analysis practices has yielded consistent results across multiple institutions. In Europe, amino acid PET has shown cost-effectiveness in distinguishing recurrent metastatic tumors from post-radiation changes in the brain, with ^18^F-FET (^18^F-Fluoro-Ethyl-Tyrosine) having the most evidence for this indication. However, there are limited data on the uptake of amino acid PET in untreated brain metastases, and uptake may vary depending on the tumor type. Recent research has demonstrated promising results, particularly indicating that ^18^F-FBPA PET with computed tomography (CT) surpasses ^18^F-fluorodeoxyglucose (^18^F-FDG) PET/CT in the differential diagnosis of malignant tumors and benign lesions [144,145]. These findings strongly suggest that employing ^18^F-FBPA PET/CT may significantly reduce the occurrence of false-positive diagnoses when compared to ^18^F-FDG PET/CT.

Hybrid PET/MRI holds promise for improving clinical outcomes by providing more precise localization of brain metastases for pre-operative planning and post-treatment monitoring [142,143]. Additionally, PET radiotracers offer potential for theranostics, where molecular imaging agents can identify and track metastatic brain disease, allowing for targeted radiotherapy. Overall, amino acid PET shows promise as a valuable tool in neuro-oncology, with its ability to provide complementary information to anatomic MRI and potential applications in both glioma and brain metastases.

However, there are still unsolved issues, particularly with targeting efficacy between cancer and healthy cells, since LAT1 is also expressed in healthy tissues. The overlapping substrate specificity should also be considered when designing novel amino-acid-based boron carriers and PET tracers.

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
