# Peer review of "Amino Acid-Based Boron Carriers in Boron Neutron Capture Therapy (BNCT)"

_pharmaceutics, 2023, doi:10.3390/pharmaceutics15122663_

Round 1

Reviewer 1 Report

Comments and Suggestions for Authors

The presented manuscript provides an extensive perspective on the development of boron carriers, particularly amino acid-based carriers, for Boron Neutron Capture Therapy (BNCT). The authors address critical issues that need to be resolved to enhance the efficiency of BNCT treatment, with a focus on the potential of large amino acid transporter 1 (LAT1) as a target for boron carriers. They also highlight the challenges faced in the development of efficient boron carriers, and the potential of amino acid-based boron carriers in the treatment of challenging tumors, particularly brain tumors, which are difficult to treat due to their location behind the blood-brain barrier (BBB). Overlall the manuscript is neat and well-organized, however, several improvements could be made to further strengthen the article.

  1. The paper does not provide a comprehensive discussion on the specific limitations of amino acid-based boron carriers in BNCT, such as their potential toxicity, tumor selectivity, and pharmacokinetic characteristics.
  2. The paper does not provide a detailed analysis of the heterogeneity of different cell lines used in BNCT research and its impact on research finding.
  3. More comparative analysis between amino acid-based boron carriers and other types of boron carriers, such as polyhedral borane clusters or boronated nanoparticles, would add depth to the discussion and provide a more comprehensive understanding of the field.

Author Response

Dear Reviewer,

We appreciate your dedicated time and valuable input in evaluating our manuscript (Manuscript ID: pharmaceutics-2703699). In response to comments, we have provided a comprehensive, point-by-point response and have included all the suggested modifications and revisions into the revised manuscript.

1. The paper does not provide a comprehensive discussion on the specific limitations of amino acid-based boron carriers in BNCT, such as their potential toxicity, tumor selectivity, and pharmacokinetic characteristics.

We thank the reviewer for pointing out that our manuscript is missing to some extent a comprehensive discussion on certain limitations of amino acid-based boron carriers in BNCT. To address these issues, we have included toxicity data in SI when available. Additionally, the following sentences have been included in chapter 3.1.

Nonetheless, the distribution and levels of LAT1 may vary according to the specific tissue in the human body. This variability may, in part, contribute to the differences in boron uptake observed among individual patients and various tumor types. In addition to variations in the quantity and distribution of LAT1 expressions, the complex genomic and histologic diversity within tumors can also contribute to the variability in substrate uptake by tumor cells.

2. The paper does not provide a detailed analysis of the heterogeneity of different cell lines used in BNCT research and its impact on research finding.

We appreciate the reviewer for highlighting the significant challenge posed by the heterogeneity of various cell lines used in BNCT research and its influence on research outcomes. Although conducting a detailed analysis of the consequences of this heterogeneity is beyond the scope of the manuscript, we have acknowledged and addressed this challenge in the last sentences of the first paragraph in the original manuscript's discussion:

Comparison of the research findings can be somewhat complex due to the utilization of various cell lines, each with its distinct characteristics. This substantial heterogeneity within the different cell lines, including those representing distinct cancer subtypes like glioblastoma, increases the complexity when attempting to draw comparisons between results. This diversity in cell lines is also evident in this review, as studies have extensively utilized numerous cell lines. The main reason for the diverse behavior of cell lines lies in the differential expression levels of genes, which in turn result in varying levels of proteins, such as amino acid transporters like LAT1.

Furthermore, we have discussed more realistic in vitro models such as 3D cell models (e.g. spheroids) in paragraph 3.4. We hope this addresses the reviewer´s concern regarding the issue.

 3. More comparative analysis between amino acid-based boron carriers and other types of boron carriers, such as polyhedral borane clusters or boronated nanoparticles, would add depth to the discussion and provide a more comprehensive understanding of the field.

We have added one paragraph to the manuscript introduction as proposed by the reviewer.

Reviewer 2 Report

Comments and Suggestions for Authors

This manuscript aimed to review the developmental journey and challenges encountered over the years in the field of amino acid-based boron delivery compound development. The review was well-written and was meaningful to the readers of this field. The comments are listed as follows:

1. The full name of all abbreviations should be mentioned when they appear for the first time in the Abstract or the Main text.

2. The tables should be revised as Three-line tables.

Author Response

Dear Reviewer,

We appreciate your dedicated time and valuable input in evaluating our manuscript (Manuscript ID: pharmaceutics-2703699). In response to comments, we have provided a comprehensive, point-by-point response and have included all the suggested modifications and revisions into the revised manuscript.

1. The full name of all abbreviations should be mentioned when they appear for the first time in the Abstract or the Main text.

Thank you for bringing this to our attention. We have now included all full names and their corresponding abbreviations in the manuscript. In case of cell line names that are abbreviations (e.g.) HEK, the cell line type has been described.

2. The tables should be revised as Three-line tables.

We appreciate the reviewer's suggestion and have revised the tables as Three-line tables in the manuscript. In the Supplementary tables, however, we havekept the original style, as they present a large number or results and horizontal lines are helpful for the reader.

Reviewer 3 Report

Comments and Suggestions for Authors

BNCT is a promising experimental modality for treating certain types of cancer. However, the lack of optimized boron carriers remains an unsolved problem. Surprisingly, it has been easier to replace nuclear reactors as a neutron source with portable devices than to find better boron carriers than BPA and BSH.

In this review, the authors summarize over six decades of research on amino acids as boron carriers for BNCT. I advise to accept this useful overlook for publication however, there are a few comments that need to be addressed:

·       At least two other classes of low molecular weight biomolecules, nucleosides, and sugars, have been studied for a long time as potential boron carriers, and the literature devoted to them is probably as extensive as that concerning amino acids. It would be helpful to briefly comment on nucleoside and sugar boron carriers in the introduction to give the topic a broader perspective.

·       Many hopes are placed on the potential of LAT as a target for amino acid boron carriers despite its already known drawbacks such as the need for amino acid boron carriers to compete with the high concentrations of natural amino acids in the blood or low tissue specificity of the LAT transporters. Nevertheless, it is an interesting concept that deserves broader treatment in this work. The authors should provide a representative example of docking studies of boronated amino acids and LAT1, along with related graphics showing preferred docking positions and a brief discussion on amino acid modifications that should be avoided, if they were to be transported by LAT1

·       The authors need to address the issue of toxicity. Although the review discusses several factors that boron carriers must meet, such as boron accumulation and cellular uptake, water solubility or specific targeting, it does not discuss toxicity, which is a crucial factor.  Perhaps all the compounds tested and described in this review were not toxic, but if so, such a statement, along with the adopted toxicity criterion, should be provided.

·       Lastly, the supplementary tables S1-3 would be more beneficial to the reader if they were supplemented with a column specifying the numerical values of accumulation wherever possible, even if in different cited works they were given in different units.

·       Additionally, the review should clearly indicate which of the described compounds were tested in in vitro or in vivo BNCT experiments.

PS My actual rating is "accept after minor revision", I chose the "reconisder after major revision" option only because according to the journal policy “a recommendation of "Minor revisions" implies that the manuscript can be accepted without the reviewer's further review”, and I believe that this policy is inappropriate, especially in the case of review work.

Author Response

Dear Reviewer,

We appreciate your dedicated time and valuable input in evaluating our manuscript (Manuscript ID: pharmaceutics-2703699). In response to comments, we have provided a comprehensive, point-by-point response and have included all the suggested modifications and revisions into the revised manuscript.

“At least two other classes of low molecular weight biomolecules, nucleosides, and sugars, have been studied for a long time as potential boron carriers, and the literature devoted to them is probably as extensive as that concerning amino acids. It would be helpful to briefly comment on nucleoside and sugar boron carriers in the introduction to give the topic a broader perspective.”

We have added one paragraph to the manuscript introduction as proposed by the reviewer.

“Many hopes are placed on the potential of LAT as a target for amino acid boron carriers despite its already known drawbacks such as the need for amino acid boron carriers to compete with the high concentrations of natural amino acids in the blood or low tissue specificity of the LAT transporters. Nevertheless, it is an interesting concept that deserves broader treatment in this work. The authors should provide a representative example of docking studies of boronated amino acids and LAT1, along with related graphics showing preferred docking positions and a brief discussion on amino acid modifications that should be avoided, if they were to be transported by LAT1”

In response to the reviewer's suggestion, we have incorporated a representative docking example (see Figure 2) into the discussion.

”Additionally, the review should clearly indicate which of the described compounds were tested in in vitro or in vivo BNCT experiments.”

If this remark is referring to irradiation experiments, then we have addressed this by including data in SI (Table S8). In general, studies involving in vivo experiments have been identified in the main text of the original manuscript.

Round 2

Reviewer 3 Report

Comments and Suggestions for Authors

All my concerns have been addressed. I recommend publication of this work.

The reviewer appreciates the authors' effort to consistently supplement boron accumulation and distribution data, in vitro and in vivo BNCT and toxicity data with numerical values in SI.